# Symptom appraisal, help-seeking and perceived barriers to healthcare seeking in Uganda: an exploratory study among women with potential symptoms of breast and cervical cancer

Amos Deogratius Mwaka,[1] Fiona M Walter  ,[2,3] Suzanne Scott,[4] Jane Harries,[5] Henry Wabinga,[1] Jennifer Moodley[5,6,7]

► Prepublication history and additional materials for this paper are available online. To view these files, please visit the journal online (http://dx.doi.org/10.1136/bmjopen-2020-041365).

For numbered affiliations see end of article.

**Correspondence to**
Dr Fiona M Walter;
fmw22@medschl.cam.ac.uk

## ABSTRACT

**Objective** We assessed the process of recognising abnormal bodily changes, interpretations and attributions, and help-seeking behaviour among community-based Ugandan women with possible symptoms of breast and cervical cancer, in order to inform health interventions aiming to promote timely detection and diagnosis of cancer.

**Design** Qualitative in-depth interviews.

**Setting** Rural and urban communities in Uganda.

**Participants** Women who participated in the African Women Awareness of CANcer cross-sectional survey who disclosed potential breast and cervical cancer symptoms were eligible; recruitment was purposive. Interviews were conducted in women's homes, lasted between 40 and 90 min, were audio-recorded, transcribed verbatim and translated to English. Thematic analysis was used to identify themes and subthemes, underpinned by the conceptual framework of the Model of Pathways to Treatment.

**Results** 23 women were interviewed: 10 had potential symptoms of breast cancer and 13 of cervical cancer. Themes regarding symptom appraisal and help-seeking included the: (1) detection and interpretation of abnormal bodily sensations; (2) lay consultations regarding bodily changes; (3) iterative process of inferring and attributing illnesses to the bodily changes; (4) restricted disclosure of symptoms to lay people due to concerns about privacy and fear of stigmatisation; (5) help-seeking from multiple sources including both traditional and biomedical health practitioners, and (6) multiple perceived barriers to help-seeking including long waiting times, lack of medicines, absenteeism of healthcare professionals, and lack of money for transport and medical bills.

**Conclusion** Women with potential symptoms of breast and cervical cancer undergo complex processes of symptom interpretation, attributing symptoms or inferring illness, and lay consultations before undertaking help-seeking and management. Increasing community understanding of breast and cervical cancer symptoms, and tackling perceived barriers to health-seeking, could lead to prompt and appropriate symptom appraisal and help-seeking, and contribute to improving cancer outcomes.

### Strengths and limitations of this study

► To our knowledge, this is the first study set in Uganda to explore the process of recognition of abnormal bodily changes, interpretations and attributions, and help-seeking behaviour among community-based women with possible symptoms of breast or cervical cancer.

► The qualitative nature of the inquiry and strong theoretical underpinning with the Model of Pathways to Treatment enabled in-depth exploration of the research question.

► Conducting interviews by female research assistants in the home of the participants allowed for freedom of expression and in-depth responses by the majority of participants.

► Self-report of symptoms and help-seeking is subject to social desirability bias especially when participants consider that their actual perceptions, beliefs and conducts could attract unfavourable comments from the research team or other people.

## BACKGROUND

In low-income and middle-income countries, including most sub-Saharan African (SSA) countries, breast and cervical cancers are the most common cancers among women. The incidence rates of cervical cancer in Eastern and Southern Africa are 40.1 and 43.1 per 100 000 women, respectively; while that of breast cancer are 29.9 and 46.2 per 100 000 women, respectively.[1] In these regions, patients with breast and cervical cancer are often diagnosed in advanced stages.[2–4] In Uganda and Ghana, 66% of patients with cervical cancer were diagnosed in advanced stages.[4 5] In Rwanda, 76% of patients with breast cancer were diagnosed in advanced stages.[3] A recent registry-based study in SSA showed that more than half (64.9%) of patients with breast cancer are diagnosed in

advanced stages and experience low survival.[6] Advanced stage cancer at diagnosis is associated with poorer survival.[7–10] In SSA, breast cancer overall relative survival rates at 1 year and 5 years were 86.1% (84.4–87.6) and 59.0% (56.3–61.6), respectively. Relative survival varied between registries but ranged from 21.6% (8.2–39.8) at 3 years in Malawi to 84.5% (70.6–93.5) in Namibia.[6]

Poor recognition of cancer symptoms by both patients and primary healthcare professionals may contribute to the advanced stage at diagnosis and poor survival of breast and cervical cancer in SSA.[11–14] Presentation at the healthcare facilities and eventual diagnosis of cancer is often long. In Ethiopia and Nigeria, 36% and 81.6% of patients with breast cancer spent more than 90 days with symptoms before seeking care with a healthcare professional.[15 16] In Ghana, use of traditional and complementary medicines (T&CMs) was associated with prolonged time to help-seeking.[17 18] Several patient factors including individual, sociocultural and health system factors interact to influence the patients' pathway to treatment for their cancer symptoms.[19 20] In general, patients' awareness of cancer symptoms has been shown to be associated with cancer survival.[21] Similarly, in South Africa, patients with breast cancer with higher education attainment and with greater awareness and knowledge of breast cancer symptoms were statistically significantly more likely to be diagnosed with early stage disease.[22] In addition, self-management, use of traditional medicines and perceived multiple barriers to medical care have been associated with longer time to healthcare seeking for breast and cervical cancer symptoms in SSA.[3 23–25] In the USA, cancer fatalism among African Americans and Hispanics was associated with advanced stage at diagnosis, inadequate uptake of treatment and less prompt healthcare seeking for cancer symptoms.[26 27] However, most of these studies have involved hypothetical help-seeking or retrospective accounts of those already diagnosed with cancer.[12 13 28] Hypothetical help-seeking reported by people without real symptoms may not be similar to real help-seeking decisions because of somatic and emotional experiences. On the other hand, retrospective recall of symptoms and decision-making processes could be enhanced or minimised because of the cancer diagnosis.[29 30] In addition, help-seeking accounts of patients already diagnosed with cancers are likely to be influenced by their cancer diagnoses and multiple interactions with healthcare professionals. This may not be the case in symptomatic women without a cancer diagnosis and in the community.

Understanding factors influencing appraisal and help-seeking decisions for women with possible symptoms of breast and cervical cancer is a critical step to underpin the development of interventions to promote prompt recognition of symptoms and healthcare help-seeking. In this study set in Uganda, we sought to understand the processes of symptom recognition, appraisal and attribution, and perceived barriers to healthcare seeking among community women with recent experiences of possible symptoms of breast and cervical cancer, regardless of whether they had sought help or been diagnosed with cancer, in order to inform the design of interventions to promote prompt symptom appraisal and healthcare help-seeking for women in SSA.

## METHODS

### Design
A qualitative in-depth interview study.

### Study setting
The study was conducted across two sites in northern Uganda: an urban site in Gulu district and a rural site in Nwoya district. Most people in northern Uganda earn their livelihood from subsistence agricultural practices; there are low levels of education and chronic intergenerational poverty in the region. Although the level of poverty has decreased since the return of peace in 2006, the region still remains the poorest in Uganda. Greater proportions of households in the region are larger in size and less likely to have two meals a day than in the rest of Uganda.[31] HIV infection has consistently remained higher in this region compared with other parts of Uganda.[32 33]

### Study population and recruitment
This interview study was part of a larger study involving a cross-sectional survey using the African Women Awareness of CANcer (AWACAN) tool.[34 35] Eligibility criteria for this study included: women aged 18 years and over, self-reported experience of symptoms suggestive of possible breast or cervical cancer within the past 3 months, who had participated in the survey, spoke English or the Luo/Acholi language, and were willing to provide written consent to participate in an interview. Participants also needed to report two or more possible symptoms of breast or cervical cancer, but not have had breast or cervical cancer diagnoses.

In the urban site, 41 potentially eligible women were identified via the survey, but 15 did not meet inclusion criteria. Fourteen women were purposively selected to include a range of symptoms and age; these and the remaining 12 women were advised to seek help for their symptoms. In the rural site, 25 AWACAN survey participants reported symptoms suggestive of breast or cervical cancer, but 11 did not meet inclusion criteria. All the remaining 14 were considered for interviews. Interviews were conducted during October 2018 in the urban site and January–February 2019 in the rural site; they continued until data saturation was reached in both localities.[36 37]

### Data collection
Two female research assistants conducted in-depth interviews in both sites. One research assistant led the interviewing process while the other operated the digital audio-recorder and took field notes, especially of nonverbal communication and events during interviews. It was necessary for the research assistants to work in

pairs to ensure their personal security as they travelled in the villages. Data collection was mainly conducted in the women's homes, usually in quiet rooms without others present, and occasionally under trees in the family compounds. The research assistants used piloted interview guides, underpinned by the concepts of the Model of Pathways to Treatment.[19 20] The interview guide included how bodily changes were first recognised, what happened immediately after detection of a symptom, who were consulted about the symptoms and the steps undertaken to seek help (see online supplemental file 1). Sociodemographic data including age, marital status, number of biological children, highest educational attainment and HIV status were also collected. Each interview lasted between 45 and 90 min, and all interviews were audio-recorded.

## Data analysis

Audio-recordings were transcribed verbatim by the research assistants. ADM read transcripts and conducted preliminary analyses during data collection in order to determine data saturation. Before full analyses, ADM reviewed 10 transcripts (five from each site) against the recordings to ensure appropriate details were captured and transcripts labelled appropriately. ADM read three transcripts in detail, developed codes and shared the codebook with the wider study team. Codes were redefined and refined as necessary. Atlas.ti V.6.1 was used to support data analysis. Coded data segments from all transcripts were aggregated and retrieved for detailed analyses. These underpinned the inductive development of themes and subthemes, focusing on the research objective and the concepts in the Model of Pathways to Treatment.

## Patient and public involvement

Patients were not involved in the planning and implementation of this study. However, the public were involved through the district and community leaders. The district health officers and district women representatives on the district councils of the two study districts were involved in the study. These local leaders were part of our Project Advisory Committee which we formed to provide advice and guidance to the study team to ensure that the study is conducted in a culturally acceptable manner, and the findings are relevant to the local context in order to inform local and national health policies on targeted interventions to promote cancer awareness and early detection. The study objectives and tools were discussed with these stakeholders. They provided guidance on how our study could be refined to best fit the local context. We therefore refined the study tools to capture aspects of women's lives that relate to information seeking and access, healthcare seeking and decision-making in the families. In addition, the local leaders will be involved in disseminations of study findings. Their presence during dissemination is expected to make the population more confident in the results. The involvement of the local decision makers potentially increases the chance that these study findings could be acceptable and put into use.

## RESULTS
### Characteristics of the participants

Twenty-three women were interviewed: 10 had possible symptoms of breast cancer and 13 had possible symptoms of cervical cancer. Two self-reported being HIV positive and on antiretroviral therapies. Overall, the median age was 31 years (range: 18–66 years; urban site 27 years (range: 22–66 years), rural site 41 years (range: 18–64 years)) (see table 1).

### Main themes

Themes regarding symptom appraisal and help-seeking intervals included the: (1) detection and interpretations of abnormal bodily sensations; (2) lay consultations regarding bodily changes; (3) iterative process of inferring and attributing illnesses to the bodily changes; (4) non-disclosure of symptoms to lay people due to concerns about privacy and fear of stigmatisation; (5) help-seeking from multiple sources including both traditional and biomedical health practitioners, and (6) multiple perceived barriers to biomedical care. Supporting quotations are presented followed by participant characteristics (age group and symptoms of cancer type).

### Detection and interpretation of abnormal bodily sensations

Detection and initial interpretation of bodily changes as normal or abnormal punctuated participants' thoughts and course of actions regarding their symptoms. The circumstances when bodily changes were first realised were critical to initial interpretations.

#### Expected bodily changes within a person's lifespan

The age of the woman was critical in deciding whether a certain bodily change could be due to an illness or normal changes. Young women could readily associate bodily changes with menstrual cycles and possibilities of pregnancies, while older women were more concerned with menopausal changes and age-related ailments.

> That problem, I thought it was my state of menopause that I reached and I stopped giving birth. I thought that way because I wasn't conceiving or giving birth anymore. (Rural_51–70_Cervical)

Participants who had ever breast fed or been pregnant initially perceived their breast symptoms to be due to usual bodily changes, and paid little attention until symptoms persisted or became painful.

> When this lump came up, I thought that probably it is as usual when it would happen when I'm breast feeding. But now I'm not even breast feeding, but the lump on my right armpit is still there and painful. (Rural_51–70_Breast)

**Table 1** Characteristics of respondents and reported symptoms

| Participants | Study site Urban | Rural | Age group (years) | Marital status | Education attainment | Occupation | Symptoms | HIV status |
|---|---|---|---|---|---|---|---|---|
| P1 | X | | 31–50 | Widow | Primary 5 | Unemployed | Pain and lump in breast | Negative |
| P2 | X | | 31–50 | Married | Primary 4 | Business/trader | Pain and lumps in both breasts | Positive |
| P3 | X | | 51–70 | Widow | Primary 6 | Business/trader | Pain and lump in breast | Negative |
| P4 | X | | 18–30 | Separated | Primary 3 | Waitress in restaurant | SVD, vaginal bleeding between periods, persistent LAP | Negative |
| P5 | X | | 18–30 | Married | Senior 4 | Business/trader | Pain and lump in breast | Negative |
| P6 | X | | 18–30 | Married | Senior 4 | Business/trader | Vaginal bleeding during/after sex, persistent LAP | Negative |
| P7 | X | | 18–30 | Married | Tertiary education | Business/trader | Pain and lump in breast | Negative |
| P8 | X | | 18–30 | Married | Senior 3 | Business/trader | SVD, persistent LAP, pain and discomfort during sex | Negative |
| P9 | X | | 31–50 | Widow | Primary 3 | Business/trader | SVD, persistent LAP | Negative |
| P10 | X | | 18–30 | Married | Primary 7 | Unemployed | Lumps in armpits and both breasts | Negative |
| P11 | X | | 51–70 | Married | Primary 6 | Unemployed | Pain and lumps in both breasts | Unknown |
| P12 | | X | 51–70 | Married | No formal education | Unemployed | Lumps in armpits and both breasts | Negative |
| P13 | | X | 31–50 | Married | Primary 2 | Unemployed | SVD, persistent LAP, pain and discomfort during sex | Positive |
| P14 | | X | 18–30 | Married | Primary 5 | Unemployed | Pain in nipple, lump in breast | Negative |
| P15 | | X | 31–50 | Married | Primary 2 | Unemployed | Lump and pain in breast, dimpling of breast skin like orange peel | Negative |
| P16 | X | | 18–30 | Separated | Primary 6 | Unemployed | SVD, persistent LAP | Negative |
| P17 | | X | 31–50 | Married | Primary 6 | Business/trader | Persistent LAP, discomfort during sex | Negative |
| P18 | | X | 31–50 | Married | No formal education | Unemployed | Persistent LAP, discomfort during sex | Negative |
| P19 | | X | 51–70 | Married | Primary 1 | Unemployed | SVD, persistent LAP, pain and discomfort during sex | Negative |
| P20 | | X | 18–30 | Married | Primary 3 | Unemployed | SVD, vaginal bleeding between periods, persistent LAP | Negative |
| P21 | | X | 31–50 | Married | Primary 3 | Unemployed | SVD, vaginal bleeding between periods, persistent LAP | Negative |
| P22 | | X | 18–30 | Married | Primary 7 | Unemployed | Persistent LAP, discomfort during sex | Negative |
| P23 | X | | 18–30 | Married | Tertiary education | Accountant | SVD, persistent LAP, itching in the vagina | Negative |

LAP, lower abdominal pain; SVD, smelly vaginal discharge.

### Chronology with recent events

Bodily changes were interpreted based on circumstances around their onset. For example, a participant who started to experience abnormal bodily changes following a medical procedure or surgery would infer the changes to the procedure.

I thought that was an effect from the contraceptive I had inserted in my arm. Before I started using that implant, those symptoms were not there; it was not happening. (Urban_18–30_Breast)

Some participants interpreted their bodily changes in respect to their medical histories including post-abortion complications and effect of birth trauma following deliveries.

I don't really know […]. I also had a miscarriage with him but my uterus wasn't 'cleaned'. The doctor said after such an incident one has to undergo 'cleaning' of the uterus in order to be healthy once again. So that is why I feel it was that problem causing the symptoms […]. I feel that probably there are some remaining dirt in my womb. (Rural_31–50_Cervical)

### Poor personal hygiene, crowded living environment and risk of infections

Living in crowded unhygienic dwellings especially rented units influenced the initial interpretations of bodily changes experienced. Women considered personal hygiene as critical determinants of gynaecological symptoms. Women who thought their personal hygiene was compromised were quick to think of hygiene-related causes or infections, especially if they did not have a sexual partner who they could suspect for transmitting a sexually transmitted disease. This view was predominantly held among younger urban women with gynaecological symptoms.

I first thought it could be candida … because it also causes itching in the vagina. I thought of the fact that there are many tenants here, so the issue of sharing the bathroom could be the problem; you could easily get it. (Urban_18–30_Cervical)

May be poor hygiene, because from school you bathe in a crowded place. You bathe where everybody shares things; you share the buckets. So I think poor hygiene was one of the factors, yeah. (Urban_18–30_Cervical)

### Lay consultations regarding bodily changes

Several factors and concerns influenced women to share their symptoms with other people, including husbands, siblings, trusted friends and sometimes parents. Participants engaged in lay consultations for various reasons.

### Seeking financial and practical support

Most participants shared their symptoms with significant others to solicit financial and practical support. These women needed to reach the healthcare facilities for care, but lacked money or means of transport.

I thought that being my husband, if I told him he would take me to the health facility. He told me that he would have taken me … but he doesn't have the money. I didn't share it with anyone else because nobody else could help me with the money. (Rural_31–50_Cervical)

### Learn and benefit from experiences of others

Many participants shared their symptoms with other people either because they trusted them or wanted to know more from them to help them decide the course of actions for their symptoms. Some of the confidants were healthcare professionals related to the participants or had had similar symptoms in the past and or were considered to know more about the symptoms.

I tried to tell my Mother in-law, and also my sister in-law who had recently returned home, then later I told my husband. I shared with her because I felt she was a mature person and could be having some ideas about it. So she just responded by saying; that definitely needed me to go to the hospital and was not to be tampered with […]. I chose her because I felt she was close to me and we were relating so well. (Rural_18–30_Breast)

They looked to be more informed that is why I also went to them. One of them told me that she also experienced those symptoms when she was pregnant but now it's not there. That she went to the health facility and was treated, so she was saying that if I also go and get treatment I will feel better. (Urban_18–30_Cervical)

### Social accountability and avoidance of suspicion from family

A few participants shared with family, especially when the woman found it difficult to accomplish her roles in the family because of her illness.

The reason as to why I decided to tell them is because when I have the pain sometimes I need to lie down for a while to rest and calm the pain. But if you don't tell them (in-laws) they may think you are just a lazy woman who doesn't want to do work, so I had to tell them so that they can be aware. (Urban_18–30_Breast)

A few participants shared their gynaecological symptoms with their husbands not just to get financial support to seek care, but rather because they wanted to avoid their husbands suspecting them of engaging in sexual intercourse with other men.

I decided to share with him (husband) because I didn't want him to suspect me of having cheated on him and got the infection. So I told him that I had never cheated on him or had these kind of symptoms before; so it was him who had infected me with it. (Rural_18–30_Cervical)

### Non-disclosure of symptoms to lay people

While most participants shared their symptoms, most were keen not to share with people they did not trust.

### Privacy concerns

Majority of participants considered symptoms of breast and cervical symptoms as private and not appropriate to share with just anyone.

By the way sharing your personal issues anyhow with anyone makes you become a fool. For me, once you are mature enough, don't go out sharing your issues anyhow. (Rural_31–50_Cervical)

### Stigma and gossiping

Some participants who did not share their symptoms were concerned about gossiping and becoming objects of public discussions.

Some people like rumour mongering. For example, they can say such a person said this, they could now be having HIV/AIDS. They can talk; that you know, this person said they are in this state and that is already rumour mongering done. So they will be circulating your name. It is because I fear lies of people [...]. That is why it is very hard to share with people any secrets, because once you do, news go all over. This is why I no longer like sharing anyhow. (Rural_51–70_Cervical)

Self-perceived stigma could be an issue among symptomatic women, especially those with cervical symptoms.

As for other people, if you start to share your issues with them, they may judge you without assessing; they will end up airing your issues to people with other funny added lies. (Rural_31–50_Cervical)

### Low expectations from lay consultations

Some participants had low expectations from lay consultations and therefore preferred formal healthcare seeking as the only and best response to illness symptoms.

I felt I should go straight to the hospital so that the doctor himself examines me; to me sharing with someone else here wouldn't yield much because they may mention something that will never be real compared with what the doctors will examine and find out. (Urban_31–50_Cervical)

### Symptom attributions: inferring illness

Participants attributed their symptoms to various disorders including infections and cancers. Symptom attributions were guided by the nature of onset, evolution and persistence of symptoms, participants' own previous experiences, family history of illnesses, response to attempted treatments, sanctioning/disapprovals from lay consultations, and messages from the radio or healthcare professionals.

### Symptoms could be due to cancer

Gynaecological symptoms that were unfamiliar and which persisted were often attributed to cervical cancer. These women expressed a lot of worries as they were inferring illness, mostly related to death and orphanhood.

I have been asking myself what this could be! It kept me worried a lot. It was worrying me; I used to say that the symptoms I'm experiencing could be cancer

and if it is, then that would be too bad. That is what was moving in my mind [...]. I just felt that cancer disease is one that can definitely lead to things such as that. I thought so because cancer is a very bad disease, it is really a bad disease ... it hurries you and you will be short lived compared to other diseases. (Urban_31–50_Cervical)

Participants who ever screened for cervical cancer considered cancer as possible explanation for their symptoms but were unsure as they had tested negative on earlier screening tests.

I have been worried about these symptoms and signs. I first thought it was cancer, but I went for cancer screening two years ago, so am still waiting for one more year to go and test again. I thought it could be cancer because cancer is the most dangerous infection, so people fear it most. (Urban_18–30_Cervical)

I also thought about cancer but I had already tested and it was negative so they had told me to test again after three months - oh no! They said three years. I gave birth in 2013, so I tested in 2014. They had told me to go back in the previous year, 2018, but I haven't gone back again ... I thought that probably it is the one starting. Because I heard that when you start experiencing smelly vaginal discharges that could be a sign of cancer. All I thought about was that it could be syphilis, if not, then cancer. (Rural_31–50_Cervical)

Severity and persistence of symptoms as well as messages from healthcare professionals informed and or reinforced participants' attributions of symptoms to possible cervical cancer.

[My] lower back/abdominal pain made me think it was cervical cancer causing these symptoms. Because the lower abdominal pain was too much. But also I heard from the health facility. They were educating those who had gone to test for that cancer. The healthcare provider said that you can get this cancer from a man who has had sex with a woman who has the virus then he can transmit it to you through sex. (Rural_18–30_Cervical)

Persistence of bodily changes beyond ordinary expectations and severity of symptoms caused participants to review their attributions, often considering more serious illnesses including cancers.

I thought probably it could be cancer because of the much pain I was experiencing on my breast, or that it was just normal pain resulting from the back pain I sometimes experience. But then sometimes I wonder why one side of my breast has a lump and the other doesn't; so all these leave me in wonder of what could be the problem. (Urban_31–50_Breast)

Participants who were aware of breast cancer symptoms immediately attributed the changes experienced in their breasts to possible breast cancer. Most reported that they

commonly learn about symptoms of illness from radio programmes.

> The reason as to why I thought it was breast cancer is because I always listen to the radio and these doctors always talk about cancer and how cancer can start and also the signs and symptoms of cancer on people. So they tell you when you see such signs, please go to hospital. (Urban_31–50_Breast)

Persistent and relentless pain not responding to commonly used medications or home-based remedies often led participants to think of more serious illnesses including cancers.

> I thought it could be cancerous because of the pain. The pain used not to cool before I went to the hospital; so I thought it was the toughest disease, cancer […]. At least HIV/AIDS, it is easy to get drugs for it, but this cancer has no drug, till you die and get buried; and that is its medicine. (Urban_51–70_Breast)

### Sexually transmitted infections and other gynaecological infections

Participants who had symptoms of vaginal itching and discharge readily attributed the cause to candidiasis, believed to be transmitted both through sexual intercourse and poor personal hygiene. Some attributed their bodily changes to sexually transmitted infections and blamed symptoms on their husbands who were often considered promiscuous.

> I thought that it could be some sexually transmitted infection. I was thinking that it could be gonorrhoea because if it were syphilis, I would have known it because it comes with a lot of pain, and I don't have the smelly vaginal discharge. (Urban_18–30_Cervical)

> I thought that probably my husband cheated on me with an infected person because I had never experienced these symptoms. […] ever since he was away for four months from October to February without coming home even one single day, then when he comes back after sometimes, I start experiencing these symptoms! So I had every reasons to be suspicious. (Rural_31–50_Cervical)

### Help-seeking for symptoms

Women sought help from both biomedical and traditional health practitioners including faith-based healers. Participants chose a particular help-seeking route depending on perceived causes, knowledge of previous similar illnesses that responded to a particular treatment approach, advice from social networks and messages from the media especially local radio health programmes.

### Healthcare help-seeking

Most participants sought care in healthcare facilities because they needed tests to confirm the causes of their health problems; they felt these would direct them to the appropriate remedies.

> I went to the doctor because I know he is the only person who can help with my condition; they have testing kits and they can confirm without doubts. They have machines that test and confirm the truth. Those things that are just guessed are not helpful at all. (Urban_31–50_Cervical)

Most sought care in healthcare facilities whenever symptoms worsened and or caused worries, or when home-based remedies did not lead to improvement in symptoms. Persistent pain was a very common trigger for healthcare seeking. The media, especially radio talk shows, encouraged a significant proportion of participants to seek care at health facilities.

> For me the first signs I saw was the swelling of my breast and I warmed water and started massaging on the swollen part but it didn't help, so I decided to go to hospital and told the doctor. Another reason as to why I visited the hospital with the symptoms of my sickness is because of the constant teaching and reminders by doctors over the radio during their radio talk shows that whenever you have any signs and symptoms which is unusual, you should go to hospital. […] I believe if you go to the hospital early enough with any sickness the doctors will find the name of the sickness and cure you if possible. (Urban_31–50_Breast)

> I went to the clinic, bought medicine and took it. But there was no improvement that's why I decided to go to the hospital. It was in March this year (2018) that I went to the clinic; but this pain started after giving birth. What made me go to the hospital was because I was worried. I was worried that I could be having a sexually transmitted disease or cervical cancer. (Urban_18–30_Cervical)

Advice from healthcare professionals when women go for antenatal and postnatal care, and immunisation for their children motivated them to seek care whenever they experienced symptoms of illnesses.

> They say it; doctors always talk when we take our children for immunization. Those responsible for the hygiene and sanitation of the hospital once you take your children for immunization or antenatal care, they often tell us that once we get to see any funny signs of any sickness, we need to rush to the nearest health facility early enough. Otherwise the symptoms will worsen, because if you feel pain in your body, you need to go to the hospital. So I took that up and rushed to the hospital because I was also feeling that in my body. (Rural_31–50_Cervical)

Participants were also motivated to seek biomedical care for concerns regarding the welfare of their children. They would want to remain alive and nurture their children their own way.

> The reason why I decided to go to the hospital that time when I was feeling the symptoms of my lower

abdomen and the waist, is because I decided that I should stay healthy … to care for my children. I should surely remain healthy and keep them. (Rural_31–50_Cervical)

Perceived inconveniences to others and personal discomfort due to symptoms were other factors that triggered healthcare seeking among participants. Symptoms of cervical cancer including smelly vaginal discharge and excessive vaginal bleeding can be very inconveniencing and restrictive to the movement of a woman.

I went because I was not comfortable among my fellow girls; you could see others are happy and for me I have my hidden problems, so I had to go and solve it with the doctor. (Urban_18–30_Cervical)

### Traditional and complementary medicines

A few participants sought care from traditional health practitioners including herbalists and faith-based healers who used prayers, sometimes combined with herbal preparations, to treat different ailments. Some participants thought T&CMs could be useful while others contended that T&CMs are fake, and that the practitioners are dishonest. Most participants thought that T&CMs do not work. They said the medicines are given based on guess work, as there are no tests done to identify cause of illness. These negative views about T&CMs were held by participants from both urban and rural sites, and with both breast and cervical symptoms.

I don't even think about it because some usually walk past here hawking herbal drugs claiming they have what treats ulcers, blood pressure, cancer and many more diseases, but I always tell them I don't have money to buy and that I don't use herbs. There is nothing that can attract me in buying local herbs. My opinion is that traditional herbs nowadays don't work or not effective. (Urban_31–50_Breast)

With traditional medicines, you can be given just anything that will not even help you. So I will never make a commitment to use traditional medicine. They may even just go and pluck the leaves of black jack then they brand it that it is medicine for such and such a disease and yet in actual sense, it will not be the cure for that disease. So most often I stick to my decision of going to the hospital because their medicines will have been tested. (Urban_31–50_Cervical)

I haven't got any medicine from any traditional healer and I personally don't want. To tell you, these traditional medicines even if you are given in regards to these symptoms in the stomach, they will not work … So I really wonder whether the traditional medicines really treat those diseases. (Rural_31–50_Cervical)

A few of the participants who admitted to using T&CMs testified that T&CMs, the ones they used, did not work.

I don't support that, and … I will stop T&CM use because I personally tried and it failed to work; it didn't help me with anything. It is true … I was instructed to go get the traditional medicine … I first went to that other T&CM practitioner, then I went to the nurse. (Urban_18–30_Breast)

I tried taking traditional medicine in vain, and when I tried to use that without success, then my husband told me that its better I try to go to the hospital instead so that it is the doctor to feed me with the right information. I used to dig out the roots then I would drink the juice. No improvement was realized. (Rural_51–70_Cervical)

On the other hand, a substantial minority of participants who had ever used a T&CM reported that they are effective and can be used alongside modern medicine. They felt that when the cause of an illness is due to malevolent spirits or breaching cultural norms, then traditional health practitioners are helpful. Participants from the rural site were more likely to report use of T&CM than those from the urban site.

When I used to pass sticky stool and also felt pain in the lower abdomen, the thing that helped me the most was one traditional medicine … it was called *tusua* (not real name). That is a type of tree … I would dig out the roots then pound it until fine, then I boil for long, then I filter it into the cup, and afterwards when it cools I take it, then it would end up the stickiness of my stool immediately. I also used to have tiny itchy rashes in between my thighs. […] I used to break the leaves of *okala* tree (not real name). You know that tree has some whitish sap, so I would harvest that and place on the spot of the blisters and the sap from *okala* ended up curing, till to date there are no rashes. (Rural_31–50_Cervical)

### Perceived barriers to healthcare help-seeking

Participants discussed several barriers to prompt healthcare help-seeking. These included health system challenges including lack of medicines, long waiting time to see healthcare professionals, lack of money and competing social obligations.

#### Lack of medicines in the public health facilities

Most participants who contemplated healthcare help-seeking for their symptoms lamented the sorry state of the public healthcare facilities. They reported that there were often no medicines available.

I just get discouraged with the thought of going there and being told to buy the recommended drugs from outside. (Urban_51–70_Breast)

They kept telling me that the drugs they need to inject me with is out of stock! I went back two times and each time the same story, so I never went back again. (Urban_18–30_Cervical)

#### Long waiting times

Due to the long queues and extended waiting times to see the healthcare professionals, especially in the public

health facilities, participants often said they would consider alternative avenues to healthcare help-seeking. This view was more common among participants with breast symptoms especially from the urban site.

> The main referral hospital also has its own problems; you can take like two days while still on the line to see the doctor. It is equivalent to someone coming all the way from Sudan and going back; that is the line to see the doctor. Whenever you are almost at the front of the line, the doctor leaves. So when you combine all the waiting with hunger and ulcers, you end up giving up. (Urban_51–70_Breast)

> I'm just planning to go in this coming February. [The] hospital is usually very crowded; if you go there you can spend a long time without getting treatment. (Rural_18–30_Cervical)

### Absence of healthcare professionals at the healthcare facilities

Participants from both urban and rural sites reported visiting healthcare facilities several times but not finding the appropriate healthcare professionals to address their concerns.

> As you know these government hospitals, services are not like for other private hospitals. So the experience I had was - the first time I went to the hospital, they recommended that I first have a scan done, but the person who was supposed to do it wasn't around so they told me to go back the next day, which I did and they told me the person supposed to do it was still not there so I should go back on Monday, (the following week). (Urban_31–50_Breast)

> I went and the healthcare provider picked the referral note and put it in the file where they had written my complaints. Then I was told that the doctor isn't there and that I should just go back on Friday, when the doctor will be around and also the ultra-scan machine will also be working, so that they can also do an ultra-scan the next Friday. (Rural_18–30_Cervical)

### Lack of money for healthcare help-seeking

Most participants reported financial constraints to seeking healthcare in the private facilities where there would be no crowding, and medicines available, and described unaffordable fees for consultations, tests and treatments.

> There is really pain, but the main reason why I didn't go is the lack of finances that opens every door. (Rural_31–50_Breast)

Participants reported that patients sometimes had to sell whatever they have to get some money to seek care in the private health facilities where they expect reasonable care.

> There is nothing else, because for the three times I visited [the private] hospital, and two times in the government hospital I have been selling goats of which they are now finished. There is nothing more

to sell, and yet I have not yet received any improvement. (Rural_31–50_Breast)

Rural participants also mentioned competing demands for the little money they got from selling off their belongings to support their medical expenses.

> So when I get money I will seek for help in the health facility but for now it's tricky because I have grown up children and grandchildren who are under my care. So getting the money to go to the health facility is a problem. And as you also know *Lexam* (not real name) hospital, all the services there require money. […] I plan to go to *Lexam* hospital because *Takila* regional referral hospital (not real name) is quite far away, *Lexam* is nearer to our home here. (Rural_51–70_Breast)

> It is the issue of lack of money that prevented me from going. […] I am struggling so that I raise some money to buy books for my children and enrol them in school then I will think of going to the hospital. I do want to seek help but the means to access that is what is still defeating me. I need, because I also love my body. It is only the issue of money and if I had it today, I would go straight to the hospital. (Rural_31–50_Cervical)

Transport money to the public health facilities was another challenge for women, especially those from the rural areas.

> I'm still looking for money to go with to the hospital. I will go when I get the money. The money is for transport. It is 12,000= (USD 3.3) one way, so I need 24 000= (USD 6.6) for transport on a boda boda (*hired motor cycle*). (Rural_18–30_Cervical)

## DISCUSSIONS
### Main findings

This study is one of the first community-based inquiries into symptom recognition, interpretation, appraisal, illness attribution, and consequent help-seeking among women with possible breast and cervical cancer symptoms in urban and rural Uganda. We found that women undergo a complex process of sense-making and help-seeking for their symptoms. The detection and interpretation of abnormal bodily sensations were interpreted based on expected bodily changes within a person's lifespan, perceived poor personal hygiene and crowded living environments with risk of infections. On realising abnormal bodily changes, most participants engaged in extensive lay consultations with relatives, friends and acquaintances mainly because they wanted to solicit financial and practical support, learn and benefit from experiences of others regarding the symptoms, and for social accountability and avoidance of suspicion from husbands/ families especially when symptoms were disabling and restricting participations in their daily chores. On the

other hand, a substantial minority of participants did not disclose their symptoms to lay colleagues mainly because of low expectations of support, concerns with privacy, avoidance of possibility of gossiping and concern with stigmatisation from the community. Medical pluralism was common, with participants often seeking help for symptoms from both biomedical healthcare facilities and T&CM providers including faith-based healers, despite many being sceptical about the effectiveness of T&CMs.

Our findings regarding recognition of abnormal bodily changes, interpretation and symptom attributions, as well as disclosure/non-disclosure of symptoms to lay people, are similar to findings from studies in other SSA countries such as Malawi and Ghana among patients with breast cancer.[13 38] The patients appraised their symptoms within the context of their life events and situations, and were guided by expected bodily changes, family history of cancer and suggestions from lay consultations. The attribution of symptoms changed over time depending on characteristics of the symptoms, including their nature, evolution, response/non-response to common medications and advice from lay people. Understanding the life context of people could inform the content, delivery and timing of health interventions to promote prompt and appropriate symptom recognition and help-seeking.

Our findings suggest that Ugandan women with vaginal symptoms often first attribute their symptoms to gynaecological infections such as candidiasis and sexually transmitted infections. Only if symptoms persisted or did not respond to medications did women sometimes attribute their symptoms to cancer. Similar symptoms for cancers and infectious diseases can offer great challenges to women especially if the illnesses have different pathways to care and treatment. In our earlier studies among women with an established diagnosis of cervical cancer in a hospital in the same region, most women attributed their vaginal symptoms of bleeding and discharge to cervical cancer and sexually transmitted infections. Most of these women sought care after long time periods and presented with advanced stage cancer.[4] In contrast, we found that women with breast symptoms who had heard of breast cancer mainly from radios and healthcare professionals during earlier visits to health facilities quickly attributed their symptoms to breast cancer and sought care accordingly. Similar findings were reported in an earlier Ugandan study about symptoms of cervical cancer.[12] This concurs with evidence from the UK, showing that awareness and knowledge of cancer symptoms are key in shaping people's attributions of symptoms and often prompt healthcare help-seeking.[39] We also found that pain, persistence of symptoms or symptoms interfering with daily routines could trigger decisions to seek help. Mild and intermittent symptoms were either ignored or interpreted as normal bodily changes, for example, due to the menopause or pregnancy, concurring with findings from a Malawi study of women with breast cancer.[13]

Women with potential breast and cervical cancer symptoms often engaged in lay consultations with their husbands, relatives and friends regarding their symptoms. These lay consultations were found to not only influence symptom attributions and illness inference, but also women's help-seeking intentions and behaviour. Participants consulted with other people to obtain financial and practical support to seek help, as well as to gain insights into the causes and best effective remedies for their illnesses. Sanctioning of initial symptom interpretations by significant others has been shown in other studies set in the UK and Australia to influence help-seeking decisions among people with possible cancer symptoms or recently diagnosed with cancer.[40 41] To avoid misleading advice when symptomatic women seek help from peers and significant others, the community as a whole need to be aware of cancer risk factors, symptoms and treatment. Therefore, public awareness campaigns to improve general awareness is important so that when women share their symptoms with other lay people, they may get appropriate help.

We found surprising negativity about T&CMs, although some continued to consult traditional healers. This contrasts with earlier Ugandan qualitative findings from asymptomatic men and women, revealing that T&CMs occupied a reasonable place in their lives mainly because the medicines were believed to work as they worked for their grandparents and ancestors, and could be useful to fill the void created by multiple barriers to biomedical care.[42] The participants in this study perceived the research assistants as healthcare providers; indeed, one was a nurse. Therefore, it is possible that the reported low opinion towards use of T&CMs could be due to social desirability bias.

Participants in this study reported several individual, interpersonal, sociocultural and health system structural barriers that delayed or prevented them from seeking healthcare. In public health facilities these barriers included lack of medicines, long waiting times and queues, and absenteeism among healthcare professionals. At a personal level, women commonly reported lack of money for transport and medical bills as their main barrier, especially in private healthcare facilities where patients pay directly for care. Most women also reported competing priorities and obligations that would hinder them from seeking healthcare. These obligations not only competed for their time but also for financial resources which were often diverted to school fees and other priorities instead of transport to health facilities and medical bills. Similar challenges to breast and cervical cancer healthcare help-seeking have been reported in other studies in SSA.[13 14 42] Insights into the perceived barriers to, and contextual factors that influence healthcare seeking from perspectives of the women can potentially inform targeted interventions to promote prompt healthcare seeking. Imposing health programmes on women without clearly understanding their perspectives and contextual factors may not have the desired impact and would be an inefficient use of resources. Health planners and policymakers therefore need to make research findings to inform their decisions.

## Limitations

Our study has some limitations. The study was preceded by a cross-sectional survey in which women were asked about potential breast and cervical cancer symptoms. It is possible that this could have influenced how women perceived and interpreted their symptoms.

Second, conducting interviews in women's homes may have influenced how some participants responded particularly in terms of the influence of family and social networks. However, we ensured that there were no non-participants at the interview venues.

Third, self-report of symptoms and help-seeking is subject to social desirability bias especially when participants may consider that their perceptions, beliefs and conduct could attract unfavourable comments from the research team or other people. However, as the research participants were assured of confidentiality, we contend that our findings accurately represent the perceptions and help-seeking behaviour of the symptomatic women included in this study.

## CONCLUSION

Symptom attributions and consequent help-seeking may be influenced by information from radios or healthcare professionals, previous experiences with similar illnesses, approval or sanctioning by lay colleagues, and the onset, nature, persistence and severity of their symptoms as well as their response to medications. We found several inter-related sociocultural and structural barriers to healthcare help-seeking. In order to promote timely help-seeking and early detection of symptomatic breast and cervical cancer, targeted public health messages and campaigns to increase knowledge of women on breast and cervical cancer symptoms and services available need to be coupled with economic empowerment of women and improvement in access to health systems.

### Author affiliations

[1]Department of Medicine, Makerere University, College of Health Sciences, Kampala, Uganda
[2]The Primary Care Unit, Department of Public Health and Primary Care, Cambridge University, Cambridge, UK
[3]University of Melbourne, Centre for Cancer Research, Faculty of Medicine, Dentistry & Health Sciences, Melbourne, Victoria, Australia
[4]Centre for Oral, Clinical and TranslationalSciences, Faculty of Dentistry, Oral and Craniofacial Sciences, King's College London, London, UK
[5]Women's Health Research Unit, University of Cape Town School of Public Health & Family Medicine, Faculty of Health Sciences, Cape Town, South Africa
[6]Cancer Research Initiative, Faculty of Health Sciences, University of Cape Town, Cape Town, South Africa
[7]South African Medical Research Council Gynaecology Cancer Research Centre, Faculty of Health Sciences, University of Cape Town, Cape Town, South Africa

**Acknowledgements** The authors wish to appreciate the research assistants and field coordinator for their thoroughness in conducting interviews in the midst of challenges including impassable roads and traversing difficult terrains on motorcycles. We are equally indebted to the research participants who freely discussed their ideas on the research questions, thereby providing us with detailed information. We also thank the local leaders for their cooperation and participation in mobilisations and guidance for security and safety of research assistants during field visits.

**Contributors** Study concept and design—JM, FMW and ADM. Acquisition of data—ADM and HW. Analysis of data—ADM, JH and SS. Interpretation of data—all authors. Drafting of the manuscript— ADM. Critical revision of the manuscript for important intellectual content—all authors. Study supervision—JM, FMW, ADM and HW. Final approval of the submitted version of the manuscript and where to submit—all authors.

**Funding** Research reported in this article was jointly supported by the Cancer Association of South Africa (CANSA), the University of Cape Town and the South African Medical Research Council with funds received from the South African National Department of Health, GlaxoSmithKline (GSK) Africa Non-Communicable Disease Open Lab (via a supporting grant project number: 023), the UK Medical Research Council, MRC (via the Newton Fund). GSK provided in-kind scientific and statistical support as part of capacity strengthening. Award/Grant number is not applicable. FMW is Director and SS is Co-Investigator of the multi-institutional CanTest Collaborative, which is funded by Cancer Research UK (C8640/A23385).

**Disclaimer** The funders had no role in study design, data collection and analysis, preparation of the manuscript, decision to publish and where to publish.

**Competing interests** None declared.

**Patient consent for publication** Not required.

**Ethics approval** This study protocol was approved by the Lacor Hospital Institutional Research Ethics Committee (LHIREC 027/11/2016). The protocol was thereafter registered and cleared by the Uganda National Council for Science and Technology (UNCST), registration number HS60ES. Additional permission to enter into the study communities was obtained from the local leaders at the district and subcounty levels. Written informed consents were obtained form study participants before interviews. Participants received a token of 25 000 (US$6.8) for light snacks and soft drinks during interviews.

**Provenance and peer review** Not commissioned; externally peer reviewed.

**Data availability statement** Study dataset can be made available to third party following reasonable request to corresponding author.

**ORCID iD**
Fiona M Walter http://orcid.org/0000-0002-7191-6476

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
