## [Reviewer comments · BMJ Open]

ARTICLE DETAILS

TITLE (PROVISIONAL)	Symptom appraisal, help seeking and perceived barriers to healthcare seeking in Uganda: An exploratory study among women with potential symptoms of breast and cervical cancer
AUTHORS	Mwaka, Amos; Walter, Fiona; Scott, Suzanne; Harries, Jane; Wabinga, Henry; Moodley, Jennifer

VERSION 1 – REVIEW

REVIEWER	Benjamin O Anderson, MD Breast Health Global Initiative (BHGI) Fred Hutchinson Cancer Research Center Seattle, Washington USA
REVIEW RETURNED	01-Aug-2020

GENERAL COMMENTS	The authors have performed an outstanding cross-sectional qualitative examination regarding the attitudes and beliefs among a cohort of Ugandan women's with symptoms of breast or cervical cancer. The goal of the study was to explore these women's self-appraisal of those symptoms as well as their health-seeking approaches in this real world scenario. The study illuminates the complex but understandable thinking behind these women's decision-making and actions. This study is among the first of its kind. Prior studies of asymptomatic women require them to project what they anticipate they would think or do based on hypothetical symptoms that they may not have actually experienced, which may not correlate well with what they would actually do in the real-world situations. Other studies of patients undergoing cancer treatment or as cancer survivors suffer from selection bias in that these examine the beliefs of women who have already undergone diagnosis and treatment and may not well reflect the thinking and behavior of women who chose not to access the healthcare system. The importance of these attitudes and beliefs cannot be overstated. Delay in diagnosis of cervical and breast cancer is among the most significant obstacles in improving women's cancer outcomes in sub-Saharan Africa (SSA). The authors have been very careful to not overstate their findings or overgeneralize to other SSA countries or global regions. Nonetheless, the findings are likely very common among women in multiple SSA countries. A particular strength of the study is that they chose to examine findings both related to breast and cervical cancer, which are the two most common cancers among women. While the medical community tends to separate the two disease processes, from a health systems access perspective, it is meaningful to look at the two together as opportunities to improve outcomes with both malignancies. The study also exposes the significant issues that these women had to consider in seeking care.
--

	They describe their concern regarding the adverse responses they might receive from their husbands, family and community at the same time that they had to identify resources to make healthcare seeking possible. The social and financial obstacles are elucidated at the same time that the health system inadequacies are exposed. The overly simplistic concept that women do not seek care out of simple lack of awareness or self-neglect is dispelled in this qualitative study. A particularly interesting finding is that the women described surprisingly negative attitudes regarding the value of traditional medicine. A question that often arises is why women with often consult with traditional healers before seeking evaluation and treatment at the hospital. While the authors acknowledge that there could be some biasing of statements by the women who knew they were being interviewed by people coming from the allopathic healthcare community, they similarly described the major access issues that obstruct women from receiving the necessary diagnostic and treatment interventions in a timely and affordable fashion.
--	--

REVIEWER	Vincent DeGennaro Innovating Health International, Port au Prince, Haiti
REVIEW RETURNED	08-Sep-2020

GENERAL COMMENTS	This is an interesting study. Well done, although some things are not clear. You mention a cross sectional cancer survey done before this. Are those results published or do you plan on publishing them separately? If not, I'd recommend that you include them here. Without knowing the population's understanding of cancer, its hard to assess if their behaviors are appropriate. Meaning that if they don't know what breast or cervical cancer is, then we can't expect them to answer properly on why they came in seeking care or if they had thought of cancer. The cohort is young for cancer. This should be considered in the weaknesses section. The symptoms listed for possible cancer are not consistent with cancer symptoms. The vast majority of breast cancer is not painful at all upon presentation, even when masses are >5cm. Vaginal itching is not a symptom of cervical cancer (or any cancer really). Smelly vaginal discharge is much, much more likely to be due to infection and very rarely to be a symptom of cervical cancer. Attributing their behaviors and actions in the setting of possible cancer diagnosis is difficult for these three reasons (unclear cancer literacy, symptoms listed not consistent with cancer, and the age of the cohort means that cancer shouldn't really be on their mind as much. Can you pull out more from the older half of the cohort? Can you talk more about symptoms that aren't signs of candidiasis? More context for the quotes, especially as it pertains to cancer, would be helpful. I like the way the study was done, and the structure of the write up.
---

REVIEWER	Prof Carol Ann Benn Netcare Milpark Breast Care Centre of Excellence University of the Witwatersrand
REVIEW RETURNED	12-Nov-2020

GENERAL COMMENTS	This is a truly wonderful study, would love to see this conducted with a larger population.
---

VERSION 1 – AUTHOR RESPONSE

Reviewer: 1

Comments to the author

The authors have performed an outstanding cross-sectional qualitative examination regarding the attitudes and beliefs among a cohort of Ugandan women's with symptoms of breast or cervical cancer. The goal of the study was to explore these women's self-appraisal of those symptoms as well as their health-seeking approaches in this real world scenario. The study illuminates the complex but understandable thinking behind these women's decision-making and actions. This study is among the first of its kind.

Prior studies of asymptomatic women require them to project what they anticipate they would think or do based on hypothetical symptoms that they may not have actually experienced, which may not correlate well with what they would actually do in the real-world situations. Other studies of patients undergoing cancer treatment or as cancer survivors suffer from selection bias in that these examine the beliefs of women who have already undergone diagnosis and treatment and may not well reflect the thinking and behavior of women who chose not to access the healthcare system.

The importance of these attitudes and beliefs cannot be overstated. Delay in diagnosis of cervical and breast cancer is among the most significant obstacles in improving women's cancer outcomes in sub-Saharan Africa (SSA). The authors have been very careful to not overstate their findings or overgeneralize to other SSA countries or global regions. Nonetheless, the findings are likely very common among women in multiple SSA countries. A particular strength of the study is that they chose to examine findings both related to breast and cervical cancer, which are the two most common cancers among women. While the medical community tends to separate the two disease processes, from a health systems access perspective, it is meaningful to look at the two together as opportunities to improve outcomes with both malignancies. The study also exposes the significant issues that these women had to consider in seeking care. They describe their concern regarding the adverse responses they might receive from their husbands, family and community at the same time that they had to identify resources to make healthcare seeking possible. The social and financial obstacles are elucidated at the same time that the health system inadequacies are exposed. The overly simplistic concept that women do not seek care out of simple lack of awareness or self-neglect is dispelled in this qualitative study.

A particularly interesting finding is that the women described surprisingly negative attitudes regarding the value of traditional medicine. A question that often arises is why women with often consult with traditional healers before seeking evaluation and treatment at the hospital. While the authors acknowledge that there could be some biasing of statements by the women who knew they were being interviewed by people coming from the allopathic healthcare community, they similarly described the major access issues that obstruct women from receiving the necessary diagnostic and treatment interventions in a timely and affordable fashion.

Thank you for these supportive comments- they are very much appreciated.

Reviewer: 2

Comments to the Author

This is an interesting study. Well done, although some things are not clear. You mention a cross sectional cancer survey done before this. Are those results published or do you plan on publishing

them separately? If not, I'd recommend that you include them here.

The paper was published last month (Moodley et al, PLoS One, 2020); the reference has been added.

Without knowing the population's understanding of cancer, it's hard to assess if their behaviours are appropriate. Meaning that if they don't know what breast or cervical cancer is, then we can't expect them to answer properly on why they came in seeking care or if they had thought of cancer.

We were not seeking to assess whether the participants' understanding of cancer or behaviour was appropriate. Instead, as stated on p6, our aim was:

'...to understand the processes of symptom recognition, appraisal and attribution, and perceived barriers to healthcare seeking among community women with recent experiences of possible symptoms of breast and cervical cancer, regardless of whether they had sought help or been diagnosed with cancer.'

The cohort is young for cancer. This should be considered in the weaknesses section.

We agree with the reviewer that breast and cervical cancer can occur in all adult women, with incidence increasing as women age. However we were not seeking to recruit a representative cohort as would be the case for a quantitative study. Instead, as this was a qualitative study, we purposively selected participants to include a range of symptoms and age groups to give us a broad range of views (see p7). Thus we do not consider this a weakness of the study.

The symptoms listed for possible cancer are not consistent with cancer symptoms. The vast majority of breast cancer is not painful at all upon presentation, even when masses are >5cm. Vaginal itching is not a symptom of cervical cancer (or any cancer really). Smelly vaginal discharge is much, much more likely to be due to infection and very rarely to be a symptom of cervical cancer. Attributing their behaviors and actions in the setting of possible cancer diagnosis is difficult for these three reasons (unclear cancer literacy, symptoms listed not consistent with cancer, and the age of the cohort means that cancer shouldn't really be on their mind as much).

We believe the reviewer is referring to Table 1's list of symptoms reported by each interviewee. Again, we believe that the reviewer may be looking for representative symptoms of breast and cervical cancer as in a quantitative study, rather than the range of possible symptoms of cancer which would prompt a woman to first seek medical care. We also note that each woman had more than one symptom. Again, we consider this to be a strength rather than a weakness of our study.

Can you pull out more from the older half of the cohort?

We interviewed 11 younger women (aged 18-30) and include 17 quotations from them in the Results section. We reported the older women in two groups: the eight interviewees aged 31-50 from whom we included 23 representative quotes, and the four interviewees aged 51-70 from whom we included 7 representative quotes. We therefore believe that the older half of the cohort are well represented in both the analysis and the results section of the paper.

Can you talk more about symptoms that aren't signs of candidiasis? More context for the quotes, especially as it pertains to cancer, would be helpful.

Thank you for this comment which we believe goes beyond the aim of this study. Just to clarify, our aim was to understand the views of women seeking help with any possible symptoms of cancer, not just alarm symptoms such as vaginal bleeding or a painless breast lump.

I like the way the study was done, and the structure of the write up.

Thank you

Reviewer: 3

Comments to the Author

This is a truly wonderful study, would love to see this conducted with a larger population.

Again, we are delighted with these supportive comments.

VERSION 2 – REVIEW

REVIEWER	Vincent DeGennaro Innovating Health International, Haiti
REVIEW RETURNED	16-Dec-2020

GENERAL COMMENTS	Great job! Love the simple design, great translations to English, the open ended questionnaire, and the use of a CAB/PAC. Can you please provide more info on how you selected the women for the surveys? "In the urban site, 41 potentially eligible women were identified via the survey, but 15 did not meet inclusion criteria. 14 women were purposively selected to include a range of symptoms and age." What happened to the other 12 women? "Coded data segments from all transcripts were aggregated and retrieved for detailed analyses." What did you do with this data? How did you analyze it? Maybe a sentence or two how this shaped the themes you focused on? "The involvement of the local decision makers potentially increases the chance that this study findings could be acceptable and put into use." Absolutely true. Provide a few citations. The portion on TCM is not discussed much in the abstract or the introduction. Same for public health system care. "Our findings suggest that Ugandan women with vaginal symptoms often first attribute their symptoms to gynaecologic infections such as candidiasis and sexually transmitted infections." These are far, far more common than cancer so this is correct for them to assume so.
---

VERSION 2 – AUTHOR RESPONSE

Reviewer 2- Comments to the Author

Great job! Love the simple design, great translations to English, the open-ended questionnaire, and the use of a CAB/PAC.

Thank you for these kind comments.

Can you please provide more info on how you selected the women for the surveys?

"In the urban site, 41 potentially eligible women were identified via the survey, but 15 did not meet inclusion criteria. 14 women were purposively selected to include a range of symptoms and age." What happened to the other 12 women?

In the main survey, the women were selected using multistage systematic random sampling to identify the households, and simple random sampling to identify the participant from the household in case there were more than one eligible woman (Moodley et al., 2020). We then purposively selected 14 of the 26 eligible participants. The other 12 women, with all the other survey participants with symptoms, were provided referral notes to the nearest hospital capable of managing their symptoms. During data collection for the survey, the women had also been encouraged to seek care for their symptoms.

14 women were purposively selected to include a range of symptoms and age; **these and the remaining 12 women were advised to seek help for their symptoms.**

"Coded data segments from all transcripts were aggregated and retrieved for detailed analyses." What did you do with this data? How did you analyze it? Maybe a sentence or two how this shaped the themes you focused on?

We used thematic analysis to identify themes and subthemes, underpinned by the conceptual framework of the Model of Pathways to Treatment. We have added to the relevant section in the Methods section:

Coded data segments from all transcripts were aggregated and retrieved for detailed analyses. **These underpinned the inductive development of themes and subthemes**, focussing on the research objective and the concepts in the Model of Pathways to Treatment.

"The involvement of the local decision makers potentially increases the chance that this study findings could be acceptable and put into use." Absolutely true. Provide a few citations.

We do not believe that this sentence in the Methods section needs citations, as we have only made a suggestion, including the words 'potentially' and 'chance', positing that involving local decision makers could increase translation of findings into practice.

The portion on TCM is not discussed much in the abstract or the introduction. Same for public health system care.

We note that the role of TCMs is included in one of the six themes outlined in the Abstract's Results section, but we agree that neither are discussed in the Abstract or Introduction as we were working to the word limit.

In the main Discussion section we agree that we have not emphasised the public health system as we feel that it is beyond the research question. However, in the Discussion we have included a paragraph discussing our findings on TCMs:

We found surprising negativity about traditional and complementary medicines, although some continued to consult traditional healers. This contrasts with earlier Ugandan qualitative findings from asymptomatic men and women, revealing that traditional and complementary medicines occupied a reasonable place in their lives mainly because the medicines were believed to work as they worked for their grandparents and ancestors, and could be useful to fill the void created by multiple barriers to biomedical care.(42) The participants in this study perceived the research assistants as healthcare providers; indeed, one was a nurse. Therefore, it is possible that the reported low opinion towards use of traditional and complementary medicines could be due to social desirability bias.

"Our findings suggest that Ugandan women with vaginal symptoms often first attribute their symptoms to gynaecologic infections such as candidiasis and sexually transmitted infections." These are far, far more common than cancer so this is correct for them to assume so.

We completely agree.